# Diet Pattern Analysis in Alzheimer’s Disease Implicates Gender Differences in Folate–B12–Homocysteine Axis on Cognitive Outcomes

**DOI:** 10.3390/nu16050733

**Published:** 2024-03-04

**Authors:** Chi-Ping Ting, Mi-Chia Ma, Hsin-I Chang, Chi-Wei Huang, Man-Chun Chou, Chiung-Chih Chang

**Affiliations:** 1Department of Nursing, Kaohsiung Chang Gung Memorial Hospital, Kaohsiung 83301, Taiwan; cipin@cgmh.org.tw; 2School of Nursing, Meiho University, Pingtung 91202, Taiwan; 3Department of Statistics, National Cheng Kung University, Tainan 704, Taiwan; mcma@mail.ncku.edu.tw; 4Department of Neurology, Cognition and Aging Center, Institute for Translational Research in Biomedicine, Kaohsiung Chang Gung Memorial Hospital, Chang Gung University College of Medicine, 123 Ta Pei Road, Kaohsiung 83301, Taiwan; homelover@gmail.com (H.-I.C.); justin1124@cgmh.org.tw (C.-W.H.); 5School of Nursing, Fooyin University, Kaohsiung 83102, Taiwan; 6School of Medicine, College of Medicine, National Sun Yat-sen University, Kaohsiung 80421, Taiwan

**Keywords:** Alzheimer’s disease, dietary pattern, gender effect, homocysteine, B12

## Abstract

Background & Aims: Low plasma B12 and folate levels or hyperhomocysteinemia are related to cognitive impairment. This study explores the relationships among diet pattern, blood folate–B12–homocysteine levels, and cognition measurement in Alzheimer’s disease (AD) while exploring whether a gender effect may exist. Methods: This cross-sectional study enrolled 592 AD patients (246 males, 346 females) and the demographic data, blood biochemical profiles, Mini-Mental State Examination (MMSE), and a Food Frequency Questionnaire (FFQ) for quantitative assessment of dietary frequency were collected. Structural Equation Modeling (SEM) was employed to explore the associations among dietary patterns, blood profiles, and cognition. A least absolute shrinkage and selection operator regression model, stratified by gender, was constructed to analyze the weighting of possible confounders. Results: Higher MMSE scores were related to higher frequencies of coffee/tea and higher educational levels, body mass index, and younger age. The SEM model revealed a direct influence of dietary frequencies (skimmed milk, thin pork, coffee/tea) and blood profiles (homocysteine, B12, and folate) on cognitive outcomes. At the same time, the influence of dietary pattern on cognition was not mediated by folate–B12–homocysteine levels. In males, a direct influence on the MMSE is attributed to B12, while in females, homocysteine is considered a more critical factor. Conclusions: Dietary patterns and blood profiles are both associated with cognitive domains in AD, and there are gender differences in the associations of dietary patterns and the levels of B12 and homocysteine. To enhance the quality of dietary care and nutritional status for individuals with dementia, our study results still require future validations with multi-center and longitudinal studies.

## 1. Introduction

Alzheimer’s disease (AD) is characterized by a progressive decline in cognitive function with a multitude of irreversible factors. Non-pharmacological treatment emphasizes intervening in reversible factors, where nutritional or dietary interventions have gained significant research attention. The Mediterranean diet, the Dietary Approaches to Stop Hypertension (DASH) diet, or the Mediterranean-DASH Intervention for Neurodegenerative Delay (MIND) diet showed evidence of significant improvements in overall or specific cognitive domains among older individuals [1,2,3,4]. In recent decades, research has shifted towards studying dietary patterns to consider interactions between nutrients, food categories, comorbidity, and possible overall synergic effects [5]. A 4-month intervention with the DASH diet pattern demonstrated enhancements in cognitive domains along with improvements in hypertension and weight loss [6]. Another three-year follow-up study involving 604 elderly participants (experimental group: 301, control group: 304) found that while participants adhering to the MIND diet exhibited slight improvements in overall cognition and body weight in the first year, there were no significant differences in cognitive function and brain imaging results over the three-year long-term follow-up period [7].

Homocysteine is an intermediary product of methionine while its metabolism requires coenzymes B6, B12, and folate. Insufficient levels of these coenzymes lead to elevated homocysteine [8,9] and increase on the risk of dementia [10,11,12]. Increased homocysteine is a strong risk factor in accelerating cognitive decline [13] or developing AD [10,11]. In non-dementia elderly, higher level of homocysteine was also associated with poor cognitive performance [14]. Early nutritional and dietary interventions can be utilized to regulate homocysteine levels, thereby reducing the risk and mitigating the symptoms of AD [8,15]. Based on these studies, whether the diet pattern may influence the cognitive outcomes via the changes of the folate–B12–homocysteine blood profiles or whether there is gender-specific effects on the interplays between folate–B12–homocysteine blood profiles and cognition remained to be explored.

In this study, we hypothesized that the differences in the dietary patterns of AD patients may lead to changes in cognitive outcomes and the effects may be mediated by the blood profiles of homocysteine, B12, and folate. As there is a female gender effect on AD susceptibility, we also explored whether the gender effect influenced the relationships among blood profiles, cognitive outcomes, and diet patterns. At the same time, gender itself may drive different dietary patterns (and thus influence homocysteine–folate–B12 pathways). As the cognitive function consisted of different domains, we also explored whether the diet pattern or nutritional blood profiles may offer prognostic values in functional domains. To gain a deeper understanding of the associations and weightings among dietary patterns, blood characteristics, and cognition, we employed Structural Equation Modeling (SEM) to compute and analyze the relationships between diet, blood parameters, and cognition. The advantage of SEM linear models lies in their ability to simultaneously investigate multiple relationships within the same model, providing information about the fit of the data to the assumed model.

## 2. Material & Methods

### 2.1. Study Design and Study Population

This study was a cross-sectional observational study that was approved by the Institutional Review Board of Chang Gung Memorial Hospital. The study patients were treated and followed up at the Cognition and Aging Center, Department of General Neurology, Kaohsiung Chang Gung Memorial Hospital from 2008 to 2023. We enrolled patients with AD who were diagnosed according to International Working Group 2 criteria [16], and further confirmed by amyloid and tau imaging (TW-ADNI: http://tadni.cgmh-mi.com/home) if the consensus panel did not agree on the diagnosis. All of the patients were in a stable condition under cholinesterase inhibitor treatment (Donepezil) from the time of diagnosis.

The exclusion criteria were a history of clinical stroke, a negative amyloid scan, a modified Hachinski ischemic score > 4, and depression. After checking the inclusion and exclusion criteria, a total of 592 patients with AD (246 males, 346 females) were included. 

### 2.2. Clinical Assessments and Blood Chemistry Analysis

After enrollment, the demographic data of each patient were recorded, including symptom onset age (year-old), years of education, gender, and body mass index (BMI) [17]. The major blood chemistry profiles included homocysteine, vitamin B12, and folate levels. Other blood profiles included hemoglobin A1C (HbA1C), high sensitivity C reactive protein (hs-CRP), creatinine, glutamic oxaloacetic transaminase (GOT), glutamic pyruvic transaminase (GPT), albumin, high density lipoprotein (HDL), very low-density lipoprotein (VLDL), low-density lipoprotein (LDL), and total cholesterol.

### 2.3. Neurobehavioral Assessments

The Mini-Mental State Examination (MMSE) was carried out by a trained neuropsychologist to reflect the general cognitive ability. For subdomain and functional assessment, we used the Mandarin version of the Everyday Cognition (E-Cog) scale [18]. The original E-Cog scale is a validated informant-rated questionnaire based on one global factor and six domain-related factors [19], that is, memory (8 questions), language (9 questions), visuospatial (7 questions), plan ability (5 questions), organization ability (6 questions), and divided attention (4 questions); higher scores indicate worse performance in that sub-item. In the Mandarin version, 3 subdomains were included [18], which were direction (7 questions), judgment (5 questions), and care (5 questions). The psychometric properties in the E-Cog scale focus on mild symptoms in everyday function and cognition, and it is suitable to evaluate early stage AD. Ratings are made on a four-point scale: 1 = better or no change compared to 10 years earlier, 2 = questionable/occasionally worse, 3 = consistently a little worse, 4 = consistently much worse. The current paper focused on the total E-Cog score and the 9 functional sub-domains. The mean score of the domains represents the sum of all completed items, divided by the number of items completed. The mean score of each E-Cog sub-domain may range from 1–4. The total scores of the E-Cog were the sum of the 9 functional sub-domains.

### 2.4. Dietary Assessment

The food frequency questionnaire (FFQ) contains 22 food groups, and it was conducted by trained interviewers to assess dietary intake in the AD patients. To avoid possible recall bias, the FFQ was recorded by the major caregiver of the patients [20]. The frequency of intake was coded as follows: (1) Never eat or eat less than once a month, (2) Eat 1–3 times a month, (3) Eat 1–2 times a week, (4) Eat 3–4 times a week, (5) Eat 5–6 times a week, (6) Eat 1–2 times a day, (7) Eat more than 3 times a day.

### 2.5. Statistical Analysis

We used the Statistical Package for the Social Sciences (SPSS Statistics for Windows, Version 17.0, SPSS Inc., Chicago, IL, USA) for statistical analyses and Analysis of Moment Structures (AMOS) Version 26.0 for SEM, accessed on 3 August 2023. Continuous variables or ordinal data were reported as mean ± standard deviation (SD). Because most of the variables were not normally distributed and the population variances of the groups were significantly different, we assessed the differences in means using the Mann–Whitney U test (two groups) or Kruskal–Wallis test (for more than three groups). The major research interest was to study the impact of diet patterns and blood profiles (mainly homocysteine, B12, and folate) on two aspects of cognitive outcomes, MMSE and E-Cog. As the variable characteristics of the two outcome measurements were different, the variable characteristics were analyzed separately. We used SEM to access the whole AD population and the least absolute shrinkage and selection operator (LASSO) regression analysis for MMSE and gender effects. 

We excluded cases pairwise and used the correlation coefficients to find the relationships among dietary pattern food category, blood profile, and cognition scores. To avoid the multicollinearity of independent variables, the LASSO regression model was used to find the independent variables for MMSE. We reported the coefficients, *p*-values, and 95% confidence intervals (C.I). 

The SEM constructed the relationships among dietary frequency, E-Cog subdomains, and blood profiles of interest (homocysteine, B12, folate). We reported the squared multiple correlation between each layer and the external variable by one minus (the error variance divided by the variance of the observed variable). The overall fit assessment indicator of the SEM model included the chi-square value, Goodness of Fit Index (GFI), Adjusted Goodness of Fit Index (AGFI), and Root Mean Square Error of Approximation (RMSEA). If the chi-square value was small and the *p*-value > 0.05, GFI > 0.9, AGFI > 0.9 and RMSEA < 0.1, the SEM was considered well fitted. The statistical significance level was at *p* < 0.05 (two-tailed).

## 3. Results

### 3.1. Demographic Data

Five hundred ninety-two patients (male = 246, female = 346) completed the study (Table 1). The female gender had an older age of onset, higher total E-Cog scores, and lower levels in years of education, BMI, and MMSE scores. Based on the E-Cog subdomain scores, higher functional impairment was found in women. The symptom speed as reported by the caregivers was not significantly different.

Among 22 food categories, the frequencies of thin pork, fat pork, processed food, entrails, fried food, and coffee/tea showed a significant gender difference (Table 2). Among these food categories showing differences, the average food consumption frequency is significantly higher in male patients.

At least 467 patients received blood tests except for Albumin, which had 361 patients. Homocysteine levels showed a positive correlation with the frequencies of thin pork and fat pork, and an inverse correlation with the frequencies of octopus, bean, and fruit. Folate levels were inversely related to fat pork and positively related to bean and fruit (Appendix A). 

Male patients had higher homocysteine and lower B12 and folate levels than female patients (male 195, female 281, Appendix A). At the same time, they also had higher creatinine, GPT, hemoglobin, and albumin and lower HDL-C and total cholesterol compared with the female group.

### 3.2. Correlation Analysis among Factors

For all patients, factors positively correlated to E-Cog total or subdomain scores (purple color) included age, homocysteine, processed food, skim milk, and sweet drinks (Appendix A). Factors inversely correlated to E-Cog total or subdomain scores (orange color) included years of education, BMI, B12, entrails, eggs, and coffee/tea. 

### 3.3. SEM for AD Patients

#### 3.3.1. Model Path Figure and Model Fit Assessment

To check the linear relations among the three layers: dietary frequency, E-Cog subdomains, and blood profile, we set the external variables of each latent variable by significantly related variables. Because there were some missing values and outliers in the external variables, we only considered the variables with complete data (*n* = 457). 

After adding measurement-error terms and model-error terms to the variables, we built the linear structural relations in Figure 1. The numbers in the upper right corner of each layer and external variable represent the squared multiple correlation. The overall fit-assessment indicator of the SEM model is very good (chi-squared value = 39.306, degree of freedom = 38, *p* value = 0.411, AGFI = 0.980, GFI = 0.989, and RMSEA = 0.009).

#### 3.3.2. Explanation of the Influence of Each Layer

The SEM model showed that dietary frequency and blood profile explained 39% of variation of cognitive function and dietary frequency explained 4% of variation of blood profile. In Figure 1. In Figure 1, dietary frequency had three significant external variables: skimmed milk (r = 0.15), thin pork (r = −0.34), coffee/tea (r = −0.39); E-Cog included five significant external variables: memory (r = 0.887), organization ability (r = 0.951), direction (r = 0.922), divided attention (r = 0.879), and judgement (r = 0.945). The blood profile had three significant external variables: homocysteine (r = −0.656), B12 (r = 0.570), folate (r = 0.380). 

(1)Influence of dietary frequency on cognitive function: dietary frequency has significant and direct impact (*p* = 0.041).(2)Influence of dietary frequency on blood profile: dietary frequency has no impact on the blood profile (*p* = 0.165).(3)Influence of blood profile on the cognitive function: blood profile has significant and direct impact on the cognitive function (*p* = 0.012).

The standardized model path coefficient estimates and t values were as follows: dietary frequency to cognitive function: 0.62 (t = 1.973, *p* = 0.041), and blood profile to cognitive function: −0.24 (t = −2.508, *p* = 0.012). 

#### 3.3.3. Gender-Related Factors Selection for MMSE Prediction

We accessed significant factors that correlated with MMSE scores, stratified by gender (Figure 2). The common factors for MMSE prediction in both male and female patients included age, education, BMI, and the use of coffee/tea. The levels of B12 were related to MMSE in the male gender while in the female gender, homocysteine levels were inversely related to MMSE. 

As there were interactions between gender and blood profiles, we further added blood profile (homocysteine, B12, folate) into the LASSO regression analysis for the prediction weighing calculations, segregated by gender (Table 3). The upper half of Table 3 showed age (*p* = 0.023), years of education (*p* = 0.025), BMI (*p* = 0.024), B12 (*p* = 0.001), and coffee/tea (*p* = 0.045) were significantly associated with MMSE score for male patients. The lower half of Table 3 showed that years of education (*p* < 0.001), BMI (*p* = 0.007), homocysteine (*p* = 0.010), and coffee/tea (*p* < 0.001) were significantly associated with the MMSE score for female patients. Notably, in blood profiles, B12 had a positive prediction of MMSE in male patients while homocysteine had an inverse prediction of MMSE in female patients. The results of the regression analysis are similar (Appendix A).

The gender effect of homocysteine and B12 on cognitive measurement was repeatedly observed in the E-Cog total and subdomain scores (Appendix A). The level of homocysteine in male patients was not related to the E-Cog total and subdomain scores but was significantly related in female patients. The relationship between B12 and E-Cog was observed in male patients.

## 4. Discussion

In patients with AD, we investigated the effect of dietary patterns and blood profiles on cognitive outcomes, with a particular focus on gender differences. There were three major findings. First, relevant research indicated that dietary factors such as folate and vitamin B12 can influence blood homocysteine levels, subsequently affecting cognitive function [3,12,21,22,23]. The results were only partially consistent with our findings. While we noticed a direct influence of dietary frequencies (skimmed milk, thin pork, coffee/tea) and blood profiles (homocysteine, B12, and folate) on cognitive outcomes, the influence of diet frequencies was not mediated by the predefined blood profiles. Second, higher frequencies of coffee/tea were related to higher MMSE scores in both genders. Other factors related to higher MMSE scores included higher educational levels, BMI, and younger age. Finally, we found that the effects of B12 and homocysteine on MMSE differ by gender. The graphical abstract is shown in Figure 3. 

In AD, the folate–B12–homocysteine axis played a significant role in cognition [24,25,26]. The relationships of folate–B12 and homocysteine levels are bidirectional [24,26,27,28]. Our study fills a critical research gap, as we found a gender-specific trait among these three blood profiles. B12 remained the most significant factor in AD cognitive-outcome prediction in male patients, and the interrelationships among B12, homocysteine, and folate were significant. In contrast, we found homocysteine played a major role in outcome prediction in female patients, and the influence of homocysteine was related to B12 and not folate. 

Another important finding of our study revealed gender differences in dietary intake frequencies and dietary patterns were directly related to cognitive outcomes. For male patients, categories such as skim milk and coffee/tea were influential, while fruit/vegetable/egg, coffee/tea, and entrails were related to cognitive outcomes in female patients. Although folate or homocysteine was found to relate to food categories, such as bean and fruit (Appendix A), the direct effect of diet and the folate–B12–homocysteine axis profiles were not established in our study. Although significant, the weighting of B12, folate, and homocysteine on the MMSE was considered as minor compared with diet pattern per se, opening another possible non-pharmacological intervention strategy. 

It is noteworthy that coffee/tea consumption emerged as a common influencing factor in the dietary patterns of both genders. The intake of coffee/tea proved beneficial for cognition, directly impacting cognitive outcomes, consistent with other reports [20,29,30]. We believe that caffeine intake from coffee and tea affects MMSE cognitive performance, although there are conflicting findings in the literature [31,32,33]. This study discovered a correlation between the frequency of food consumption and cognitive function but did not account for intake dosage and adherence duration. Further research is needed to establish more precise neuroprotective dosage, frequency, and duration considerations.

In our SEM model, dietary frequency in AD has no direct impact on the predefined blood profiles (folate, B12, and homocysteine). The finding does not imply a lack of the influence of diet on the folate–B12–homocysteine axis. We found significant correlations between food categories and variable blood profiles (Appendix A) such as hemoglobin or lipid profiles. The relationships between food categories and homocysteine and folate levels were also established. The use of nutritional vitamins can reduce homocysteine blood markers and increase the levels of vitamin B12 and folate [34]. It is possible that the intake of trace nutrients from food has a limited impact on homocysteine, vitamin B12, and folate levels. It is also possible that the FFQ lacks the sensitivity to detect a single nutrient in a diet pattern. Because dietary frequency and E-Cog subdomains come from a questionnaire survey, errors and variances in data are usually larger. Thus, the proportion of explained variation is expected to be low in the SEM.

Our study has several limitations. Firstly, the sample was drawn from a single medical center, and it was a cross-sectional study involving questionnaire-based dietary assessments. While our study was carried out in a single medical center, the study’s results may provide methodological consistency at the cost of limited generalizability. A further multi-center study is needed to enhance and validate the results presented herein. Based on a cross-sectional design, the relationships among dietary patterns, targeted blood profiles (B12–folate–homocysteine), and cognitive outcomes by SEM only imply an association rather than a causal relationship or an upstream–downstream cascade. A longitudinal study design may help to disentangle the complex interplays among these factors. Secondly, the FFQ included the dietary frequency of each food category, but it did not measure the quantity or record changes in diet preferences. The nutrition as provided by the food may be confounded by intake amount, the medication that the patients used, or the interactions between them. A larger sample size is needed to include more confounding factors for a better understanding. We used the FFQ because it is more accessible and easier to apply in data collection. Based on a self-reported questionnaire, it is possible that a recall bias still existed in the caregivers of the patients. Additionally, dietary patterns may change over time, and using a single assessment may not capture long-term dietary habits accurately. In the future, we recommend that an objective study design using the diet photoimaging information system may help to more accurately estimate diet quantities and nutritional components. Thirdly, the dietary category survey included 22 types of food, which may not fully cover the diversity of all food choices. At the same time, the intake frequency still relied on caregivers’ recall memory, which may introduce variable recall bias. To avoid possible mistakes, we administered the questionnaires to the major caregiver, and we collected the dietary patterns, blood profiles, and cognitive tests in a short timeframe. We anticipated that the results may reflect the real-world situation. 

## 5. Conclusions

In conclusion, dietary patterns, homocysteine, B12, and folate showed direct impacts on cognition outcomes in AD, and gender-associated differences in blood and diet pattern were found. Variations in vitamin B12 and folate levels in the blood were found to have a direct or indirect impact on blood homocysteine levels, subsequently affecting cognitive function. Understanding the dietary sources of these protective nutrients for AD is crucial, as it aids in the development of personalized nutrition plans and treatments for primary prevention. This, in turn, contributes to improving the quality of life and care for individuals with AD. Further research is needed to validate the effectiveness and feasibility of implementing such strategies in clinical practice.

## Figures and Tables

**Figure 1 nutrients-16-00733-f001:**
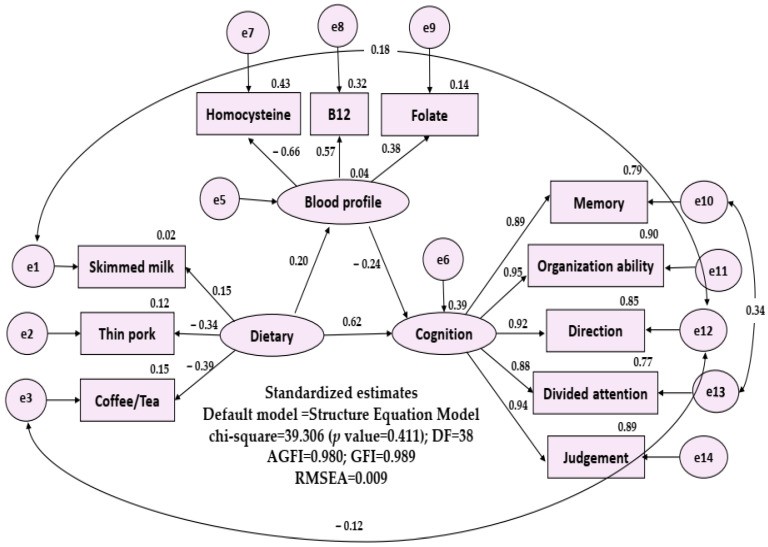
Structual equation model of possible layers (first layer: important diet pattern; second layer: b12, folate, homocysteine; third layer: everyday cognition, five subdomains). DF: degree of freedom; GFI: Goodness of Fit Index, AGFI: Adjusted Goodness of Fit Index, RMSEA: Root Mean Square Error of Approximation.

**Figure 2 nutrients-16-00733-f002:**
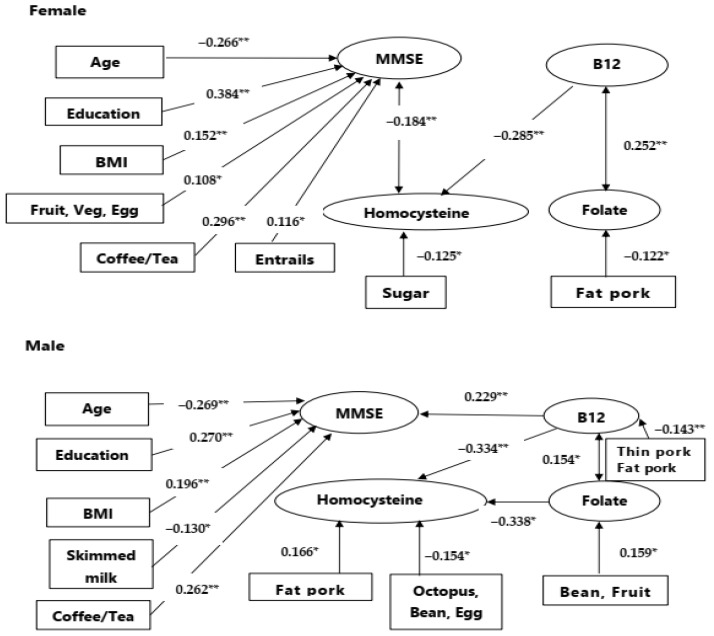
Gender effect on cognitive prediction. Upper: In male patients, B12 had significant relationships with mini-mental status examination (MMSE) score. Lower: In female patients, homocysteine had significant relationships with MMSE scores. The numbers indicated correlation coefficients. BMI: body mass index; Veg.: vegetables. Numbers indicate correlation coefficients. If there are more than two foods, the number represents the smallest correlation. * *p* < 0.05, ** *p* < 0.01.

**Figure 3 nutrients-16-00733-f003:**
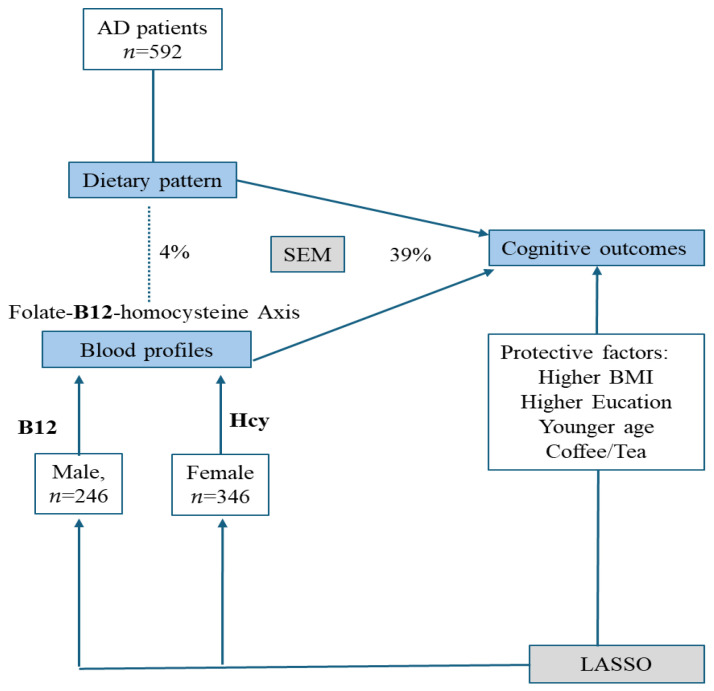
Graphical abstract of the study. The major cognitive outcomes in this study included everyday cognition and Mini-Mental Status Examinations. In all patients with Alzheimer disease, dietary pattern and blood profiles contributed significantly to the outcomes, while the blood profiles of B12–homocysteine (Hcy) showed gender effects. The solid line indicates significant relationships while the dash line showed no statistical significance. AD: Alzheimer disease; LASSO: least absolute shrinkage and selection operator; SEM: Structural equation model; BMI: body mass index.

**Table 1 nutrients-16-00733-t001:** Demographic data of patients with Alzheimer disease.

	Male (*n* = 246)	Female (*n* = 346)	MW U Test
	Mean ± SD	Mean ± SD	*p*-Value
Age at study	71.87 ± 9.00	73.27 ± 7.87	0.026 *
Age of onset	67.49 ± 9.13	69.38 ± 8.20	0.029 *
Educational year	9.86 ± 4.44	6.76 ± 4.55	<0.001 ***
Body mass index (kg/m^2^)	24.46 ± 3.53	23.35 ± 3.44	<0.001 ***
Mini-mental state examination	21.74 ± 6.96	18.88 ± 7.89	<0.001 ***
Symptom speed	3.38 ± 1.23	3.24 ± 1.30	0.219
Every Cognition (E-Cog)			
total	17.61 ± 8.28	19.02 ± 8.23	0.009 **
memory	2.37 ± 1.01	2.59 ± 0.99	0.007 **
language	1.83 ± 0.97	1.97 ± 0.98	0.029 *
visuospatial	1.84 ± 1.04	2.01 ± 1.08	0.025 *
plan ability	2.10 ± 1.06	2.27 ± 1.03	0.036 *
organization ability	2.01 ± 1.12	2.20 ± 1.11	0.013 *
divided attention	2.38 ± 1.02	2.58 ± 1.03	0.034 *
direction	1.79 ± 0.92	1.93 ± 0.90	0.009 **
judgement	1.84 ± 0.98	2.02 ± 0.99	0.004 **
care	1.42 ± 0.89	1.45 ± 0.93	0.853

SD: standard deviation; Comparisons between male and female gender using Mann–Whitney (MW) U test. Symptom speed was derived from every cognition questionnaire by informants with lower score indicating more rapid progression. * *p* < 0.05, ** *p* < 0.01, *** *p* < 0.001.

**Table 2 nutrients-16-00733-t002:** Differences of 22 food groups in Alzheimer’s disease, stratified by gender.

	Male (*n* = 246)	Female (*n* = 346)	MW U Test
	Mean ± SD	Mean ± SD	*p*-Value
**Fried food**	1.72 ± 0.91	1.60 ± 0.96	0.012 *
**Coffee/Tea**	3.44 ± 2.17	2.70 ± 2.07	<0.001 ***
**Thin Pork**	3.50 ± 1.25	3.24 ± 1.30	0.040 *
**Fat Pork**	2.65 ± 1.32	2.20 ± 1.28	<0.001 ***
**Processed Food**	1.71 ± 0.91	1.55 ± 0.78	0.031 *
**Entrails**	1.36 ± 0.62	1.23 ± 0.52	0.004 **
Oyster	1.96 ± 0.92	1.88 ± 0.86	0.276
Octopus	1.89 ± 0.85	1.82 ± 0.81	0.36
Soy Products	2.02 ± 1.03	2.12 ± 1.21	0.564
Beans	2.60 ± 1.35	2.73 ± 1.49	0.459
Full-fat milk	2.71 ± 2.03	2.77 ± 2.09	0.941
Low-fat milk	1.42 ± 1.05	1.66 ± 1.51	0.536
Skimmed milk	1.56 ± 1.35	1.65 ± 1.57	0.478
Egg	4.63 ± 1.40	4.49 ± 1.44	0.261
Vegetable	5.67 ± 0.91	5.79 ± 0.77	0.094
Mushroom	2.61 ± 1.20	2.71 ± 1.22	0.291
Fruit	5.31 ± 1.32	5.35 ± 1.24	0.95
Dessert	2.32 ± 1.46	2.26 ± 1.33	0.979
Sweet drink	1.57 ± 1.20	1.61 ± 1.21	0.439
Sugar	1.46 ± 1.00	1.49 ± 0.99	0.533
Fish	4.16 ± 1.53	4.13 ± 1.59	0.857
Chicken	3.27 ± 1.14	3.15 ± 1.24	0.135

SD: standard deviation; Comparisons between male and female gender using Mann–Whitney (MW) U test. Bold text in food group indicates items with significant differences between male and female gender, * *p* < 0.05, ** *p <* 0.01, *** *p* < 0.001.

**Table 3 nutrients-16-00733-t003:** The LASSO regression analysis for MMSE score in patients with Alzheimer’s disease.

	Unstandardized					
	Coefficients		z	*p* Value	95% Confidence Interval	
	B	Std. Error			Lower Bound	Upper Bound
All patients (*n* = 449, AIC = 2992.8)						
(Constant)	15.013	4.127	3.638	<0.001	6.926	23.101
Age	−0.120	0.043	−2.790	0.005 **	−0.204	−0.036
Education	0.444	0.077	5.749	<0.001 ***	0.293	0.595
Body mass index	0.324	0.094	3.441	<0.001 ***	0.140	0.509
Homocysteine	−0.073	0.066	−1.102	0.271	−0.204	0.057
B12	0.001	0.001	2.537	0.011 *	<0.001	0.002
Coffee/tea	0.589	0.160	3.669	<0.001 ***	0.274	0.903
Male patients (*n* = 182, AIC = 1196)						
(Constant)	17.312	6.255	2.768	0.006	5.053	29.572
Age	−0.130	0.057	−2.276	0.023 *	−0.242	−0.018
Education	0.264	0.118	2.239	0.025 *	0.033	0.495
Body mass index	0.295	0.131	2.255	0.024 *	0.039	0.552
B12	0.003	<0.001	3.236	0.001 **	0.001	0.004
Coffee/tea	0.425	0.213	2.001	0.045 *	0.009	0.842
Female patients (*n* = 269, AIC = 1824.7)						
(Constant)	14.228	5.485	2.594	0.009	3.479	24.978
Age	−0.090	0.060	−1.498	0.134	−0.207	0.028
Education	0.527	0.107	4.901	<0.001 ***	0.316	0.737
Body mass index	0.353	0.131	2.704	0.007 **	0.097	0.609
Homocysteine	−0.281	0.109	−2.594	0.010 *	−0.494	−0.069
Coffee/tea	0.783	0.223	3.505	<0.001 ***	0.345	1.221

LASSO: least absolute shrinkage and selection operator, MMSE: Mini-Mental State Examinations, AIC: Akaike information criterion, * *p* < 0.05, ** *p* < 0.01, *** *p* < 0.001.

## Data Availability

Data is contained within the article (and Appendix A).

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
