# Peer review of "Diet Pattern Analysis in Alzheimer’s Disease Implicates Gender Differences in Folate–B12–Homocysteine Axis on Cognitive Outcomes"

_nutrients, 2024, doi:10.3390/nu16050733_

Round 1

Reviewer 1 Report

Comments and Suggestions for Authors

I have enjoyed reading the manuscript entitled: Diet Pattern Analysis in Alzheimer Disease Implicates Gender Differences in Folate-B12-Homocysteine Axis on Cognitive Outcomes by Ting et al. The topic is interesting and the findings presented contribute to our understanding of the relationships between dietary pattern, blood folate-B12-homocysteine ​​levels and measures of cognition in Alzheimer's disease (AD) and whether there may be a gender effect; offering the perspective of a promising paper.

Overall, this is a good article. The writing and presentation are scientifically sound. The methodologies are appropriate and aligned with the proposed objectives. The conclusions are consistent with the evidence and arguments presented. Very well-chosen statistical analysis methods.

However, I have the following suggestions related to the improvements that should be added:

-       please summarize the main theme of the article in a graphical abstract.

-       please provide a caption for the images on page 6.

-       the References list contains several incomplete information on certain bibliographic resources, generally the lack of inclusion of pages or at the level of including authors. I suggest completing them.

-       there are spelling, punctuation and some grammar issues. Sometimes the spaces between the words are missing, other times there are too many. This will apply to the whole manuscript.

Author Response

I have enjoyed reading the manuscript entitled: Diet Pattern Analysis in Alzheimer Disease Implicates Gender Differences in Folate-B12-Homocysteine Axis on Cognitive Outcomes by Ting et al. The topic is interesting, and the findings presented contribute to our understanding of the relationships between dietary pattern, blood folate-B12-homocysteine ​​levels and measures of cognition in Alzheimer's disease (AD) and whether there may be a gender effect; offering the perspective of a promising paper. Overall, this is a good article. The writing and presentation are scientifically sound. The methodologies are appropriate and aligned with the proposed objectives. The conclusions are consistent with the evidence and arguments presented. Very well-chosen statistical analysis methods.

However, I have the following suggestions related to the improvements that should be added:

  1. 1. Please summarize the main theme of the article in a graphical abstract.

Response: We have summarized the main theme of the article in a graphical abstract as shown in Figure 3.

  1. please provide a caption for the images on page 6

Response:We have included figure legends for each figure in the revision. Thank you.

3: References list contains several incomplete information on certain bibliographic resources,

Response:We have thoroughly reviewed the sources and complete information of our references, with modifications made on reference 1, 10, 15, 29 (page 12-13 in red text)  

4:Have spelling, punctuation and some grammar issues. Sometimes the spaces between the words are missing, other times there are too many.

Response:The related errors have been checked and the edited portion has been edited by native English speaking professional for proofreading. We also attached the certificate here.

Reviewer 2 Report

Comments and Suggestions for Authors

Dear Author,

I appreciate the opportunity to review your manuscript titled "Diet Pattern Analysis in Alzheimer Disease Implicates Gender Differences in Folate-B12-Homocysteine Axis on Cognitive Outcomes." Your research significantly contributes to understanding the intricate relationships between dietary patterns, blood nutrient levels, and cognitive outcomes in Alzheimer's Disease, with an innovative emphasis on gender differences. The overall structure and execution of your manuscript are commendably well-organized, guiding the reader through your research journey effectively.

While the manuscript is fundamentally strong, I have identified a few areas where enhancements could further clarify and enrich your presentation:

Figures and Legends: The inclusion of figures throughout the manuscript is crucial for illustrating your findings. However, it seems that some figures may lack detailed legends or descriptions, which are essential for the reader's comprehension of the data presented. I recommend a thorough review of all figures to ensure that each is accompanied by a comprehensive legend, accurately and sufficiently descriptive. This adjustment will significantly aid the reader in understanding and interpreting your key results.

Single-Center Study Consideration: The design of your study as a single-center observational study at Chang Gung Memorial Hospital provides focused insights within a specific context. While this approach offers methodological consistency, it may also limit the generalizability of your findings. Acknowledging this limitation and discussing potential avenues for future multicenter studies to validate and expand upon your findings would present a balanced perspective and suggest fruitful directions for further research.

Terminology Consistency and Precision: The manuscript utilizes a range of specialized terms related to Alzheimer's Disease, dietary patterns, and cognitive assessments. Ensuring the consistent and precise use of these terms not only enhances readability but also aligns with best practices in scientific writing. Providing definitions or a section that clarifies key terms could improve the manuscript's accessibility to a wider audience, including those less familiar with specific terminologies.

With these suggestions implemented, I believe your manuscript will offer even greater insights and clarity. The careful refinement of figure legends, acknowledgment of the study design's limitations, and enhanced consistency in terminology use will all contribute to strengthening the impact of your work.

Thank you for considering these suggestions. I look forward to seeing the advancements in your research and its contribution to the field.

Best regards,

Comments on the Quality of English Language

The quality of English in the manuscript is generally good, making the research understandable. However, as someone who also struggles with English, I noticed a few areas where the language could be polished for better clarity. Simplifying complex sentences and ensuring consistency in terminology would be beneficial. Minor proofreading by a native speaker could further enhance the manuscript's readability.

Author Response

I appreciate the opportunity to review your manuscript titled "Diet Pattern Analysis in Alzheimer Disease Implicates Gender Differences in Folate-B12-Homocysteine Axis on Cognitive Outcomes." Your research significantly contributes to understanding the intricate relationships between dietary patterns, blood nutrient levels, and cognitive outcomes in Alzheimer's Disease, with an innovative emphasis on gender differences. The overall structure and execution of your manuscript are commendably well-organized, guiding the reader through your research journey effectively.

While the manuscript is fundamentally strong, I have identified a few areas where enhancements could further clarify and enrich your presentation:

  1. Figures and Legends: The inclusion of figures throughout the manuscript is crucial for illustrating your findings. However, it seems that some figures may lack detailed legends or descriptions, which are essential for the reader's comprehension of the data presented. I recommend a thorough review of all figures to ensure that each is accompanied by a comprehensive legend, accurately and sufficiently descriptive. This adjustment will significantly aid the reader in understanding and interpreting your key results.

Response: The original figures with related legends were included for better understanding. We have added a graphical abstract as Figure 3 with figure legend. Thank you.

  1. Single-Center Study Consideration: The design of your study as a single-center observational study at Chang Gung Memorial Hospital provides focused insights within a specific context. While this approach offers methodological consistency, it may also limit the generalizability of your findings. Acknowledging this limitation and discussing potential avenues for future multicenter studies to validate and expand upon your findings would present a balanced perspective and suggest fruitful directions for further research.

Response: Thank you for the reviewer’s suggestions. We have included the suggestion in the limitation section as follows (page 11 at line 350-357)

  1. Terminology Consistency and Precision: The manuscript utilizes a range of specialized terms related to Alzheimer's Disease, dietary patterns, and cognitive assessments. Ensuring the consistent and precise use of these terms not only enhances readability but also aligns with best practices in scientific writing. Providing definitions or a section that clarifies key terms could improve the manuscript's accessibility to a wider audience, including those less familiar with specific terminologies.

Response: In this revision, we have complied with the best practices in scientific writing to make the specific term consistent. Thank you for the suggestions.

With these suggestions implemented, I believe your manuscript will offer even greater insights and clarity. The careful refinement of figure legends, acknowledgment of the study design's limitations, and enhanced consistency in terminology use will all contribute to strengthening the impact of your work.

Thank you for considering these suggestions. I look forward to seeing the advancements in your research and its contribution to the field.

Response: We thank the reviewer for the suggestions and comments.

Reviewer 3 Report

Comments and Suggestions for Authors

The cross-sectional design of the study limits the establishment of causality, and longitudinal studies are needed to better understand the temporal relationships between dietary patterns, blood characteristics, and cognitive outcomes in Alzheimer's disease patients.are warranted to confirm these findings. Cross-sectional studies can only provide associations at a single point in time and cannot determine the direction of causality. Please justify it.

While SEM is a powerful tool for exploring complex relationships, it relies on the assumption of causal relationships between variables. However, without longitudinal data or experimental manipulation, it is challenging to confirm causality.

Additionally, the study did not account for potential confounding variables such as medication use, which could influence both dietary patterns and cognitive outcomes. Moreover, the reliance on self-reported dietary data introduces the possibility of recall bias, and dietary patterns may have changed over time, affecting the accuracy of the assessment.

Further research incorporating objective measures of dietary intake and exploring potential interactions with medication use is needed to provide a more comprehensive understanding of the relationship between diet, blood characteristics, and cognition in AD patients.

Additionally, the study's exclusion criteria and sample characteristics may limit the generalizability of the results to broader populations. Please make some statement in the conclusion

The reliance on self-reported dietary data via a Food Frequency Questionnaire introduces the possibility of recall bias and inaccuracies in reporting. Additionally, dietary patterns may change over time, and using a single assessment may not capture long-term dietary habits accurately. Make some satisfactory statement.

Additionally, further research is needed to validate the effectiveness and feasibility of implementing such strategies in clinical practice.

Author Response

  1. The cross-sectional design of the study limits the establishment of causality, and longitudinal studies are needed to better understand the temporal relationships between dietary patterns, blood characteristics, and cognitive outcomes in Alzheimer's disease patients are warranted to confirm these findings. Cross-sectional studies can only provide associations at a single point in time and cannot determine the direction of causality. Please justify it.

Response: Thank you for the reviewer to point out the importance of longitudinal study design and the limitation of the cross-sectional study design. Our cross-sectional study design can only imply associations among dietary patterns, targeted blood profiles (B12-folate-homocysteine) and cognitive outcomes. Based on the SEM model, a relationship between dietary pattern and cognitive measurement was found. We understand that the cross-sectional design study may limit the establishment of causality and we have included an explanation in the limitation section for the reader (page 11 at line 350-357)

  1. While SEM is a powerful tool for exploring complex relationships, it relies on the assumption of causal relationships between variables. However, without longitudinal data or experimental manipulation, it is challenging to confirm causality.

Response: By the consultation of a statistician and the literature search, SEM is suited to the management of cross-sectional data, test non-straightforward relationships and for inferential purposes [Fam Pract. 2019 Jun; 36(3): 297–303]. The model has enabled simultaneous fit of several multiple linear regressions with variables presented be either observable or latent.

We investigated the dietary pattern by caregiver’s statement, measured the blood profiles of the patients and the outcome measurements was the E-cog scores. In our SEM design, we construct the layers based on the variable characteristics (page 2, line 85-91) and investigated the associations among layers and variables, rather than on the causality. The SEM results have been included in Figure 1. In the limitation, we have included that the SEM results do not imply a causal relationship but an association among factors to avoid the confusion. We also mentioned the limitation of a cross-sectional study and the requirement of a longitudinal study design (page 11 at line 352-357).

  1. Additionally, the study did not account for potential confounding variables such as medication use, which could influence both dietary patterns and cognitive outcomes. Moreover, the reliance on self-reported dietary data introduces the possibility of recall bias, and dietary patterns may have changed over time, affecting the accuracy of the assessment.

Response: For our inclusion criteria, we mentioned that all patients received cholinesterase inhibitor treatment (Donepezil) from the time of diagnosis. Other than Donepezil, we did not set the inclusion or exclusion criteria for medication use. As the reviewer pointed out, the multi-pharmacy conditions in patients with AD may influence both dietary pattern, dietary frequency, or diet preferences. The multi-pharmacy conditions also may interfere with the cognitive performances of the patients during the test. To improve the study, the record of confounding medication should be included. However, the inclusion of more influential factors requires more sample sizes that is not possible at the present stage. page 11-12 at line 358-362

It is possible to have recall bias of FFQ based on the self-reported dietary data. Therefore, the questionnaire was applied to the caregiver but not the patients. FFQ did not include the amount of food, but only registered dietary pattern (type of food) and frequencies of food items. While it is more convenient, the FFQ only covered a portion of the dietary pattern but not record the changes of diet preferences or the intake amount. We mentioned in the limitation that a diet photo information system is more accurate to estimate quantities and analyze the nutritional components of food. Response in page 11-12 at line 365-3669

  1. Further research incorporating objective measures of dietary intake and exploring potential interactions with medication use is needed to provide a more comprehensive understanding of the relationship between diet, blood characteristics, and cognition in AD patients.

Response: The response has been included in the limitation in page 11-12 at line 354-362

  1. Additionally, the study's exclusion criteria and sample characteristics may limit the generalizability of the results to broader populations. Please make some statement in the conclusion

Response: The response has been included in the abstract and limitation in page 1 (line40-41)、page 11(line 352-354,362-363)

  1. The reliance on self-reported dietary data via a Food Frequency Questionnaire introduces the possibility of recall bias and inaccuracies in reporting. Additionally, dietary patterns may change over time, and using a single assessment may not capture long-term dietary habits accurately. Make some satisfactory statement.

Response: The response has been included (page 12 at line 370-374).

  1. Additionally, further research is needed to validate the effectiveness and feasibility of implementing such strategies in clinical practice.

Response: The response has been included in the Abstract (page 1 at line40-41) and the conclusion (page 12, line 382-383).

Reviewer 4 Report

Comments and Suggestions for Authors

The article is very well written, rigorous in method and the topic is very clinically relevant.

Some suggestions:

1. Is the column on the total of the two groups compared necessary in tables 1 and 2? 

2. Tables 1 and 2 could be more readable if the Mean and SD are inserted in a single column (M±SD)

Author Response

  1. Is the column on the total of the two groups compared necessary in tables 1 and 2? 

Response: We have removed the data of total cases and only included data of female and male group in Table 1 and 2. (page 4 Table 1, Page 5 Table 2).

  1. Tables 1 and 2 could be more readable if the Mean and SD are inserted in a single column (M±SD)

Response: In Table 1 and 2, we presented the data as Mean ± standard deviation according to the reviewer’s suggestions. Thank you.
